# Contributions of a *"Brazilianized"* Radical Behaviorist Theory of Subjectivity to the Feminist Debate on Women

Carolina Laurenti 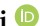

Department of Psychology, State University of Maringá, Maringá 87020-900, Brazil; claurenti@uem.br

**Abstract:** An essentialist view of gender and an individualistic concept of subjectivity have distanced psychological theories from emancipatory feminist projects. In Brazil, similar to some other psychological orientations, the behavior-analytic field has sought an interface with feminism. The anti-essentialist vein of radical behaviorism underpins the early movement toward feminism. This essay aims to expand the area of contact with feminism through a theoretical proposal for understanding women's subjectivity inspired by Brazilian behavior-analytic literature. From a contextualized, multidimensional, pluralized, and politicized view of subjectivity, women's subjectivation is forged in a tripartite complex of body, person, and "self", whose relative unity is susceptible to changes and conflicts. In a patriarchal, racist, and cis-heteronormative society, such as the Brazilian one, subjectivation is also an oppressive process. Nevertheless, the essay demonstrates that women's subjectivation can be a process of emancipatory liberation. This possibility is glimpsed within a virtuous dialectical circuit between disruptive verbal communities (uncommitted to institutional, hierarchical, and oppressive social control) and subversive subjectivities. Thus, behavior-analytic psychology has theoretical tools to situate the process of women's subjectivation not as a locus of depoliticization but as a crucial ally in constructing a more equitable and just society, as envisioned by feminism.

**Keywords:** subjectivity; gender; equity; psychology; feminism; radical behaviorism; Brazilian behavior-analytic studies

## 1. Introduction

The relationship between psychology and feminism is thorny, oscillating between attempts to disentangle these fields and efforts to restore their potential synergy (Burman 1998; Chrisler and McHugh 2018; Rutherford 2012). One of the aspects that illustrate the tension between these domains is psychology's epistemological emphasis on the individual and the notion of subjectivity derived from it. Depending on how psychology conceives of subjectivity, feminism may find an epistemological and political ally or antagonist in psychology. Pickren (2018), for example, argues that psychology has endorsed an individualistic, consumerist, and entrepreneurial concept of subjectivity aligned with neoliberal values. Based on conceptions like that, psychology can shift feminism from a political ideology to a "lifestyle" (Hooks [2000] 2015), and the process of women's subjectivation would be an individualistic quest to be a free, powerful, and successful woman (Rutherford 2018).

Another point of tension is the concept of gender. Psychology has also been accused of sustaining an essentialized concept of gender, contributing to the idea of human nature and pathologization of the feminine condition (Neves and Nogueira 2003; Prehn and Hüning 2005). In her iconic review essay, Weisstein ([1969] 1993) had already demonstrated the essentialist view of women in psychological theories and research at the time, according to which women would be intrinsically "inconsistent, emotionally unstable, lacking in a strong conscience or superego, weaker, 'nurturant' rather than productive, 'intuitive' rather than intelligent" (Weisstein [1969] 1993, p. 207).

The idea of a feminine nature is antithetical to feminist conceptions that understand women as an open and plural category historically and socially defined (e.g., Butler [1990] 1999; Hooks [2000] 2015). The alliance between an individualistic notion of subjectivity and feminine essence would move psychology away from emancipatory feminist proposals. Nonetheless, psychology is an epistemologically diverse arena, and there are efforts from several psychological theories to construct an interface with feminism (Rutherford and Pettit 2015).

In Brazil, specifically, a movement toward feminism has also been made by different psychological theories (e.g., social psychology, psychoanalysis) (Nuernberg et al. 2011; Oliveira and Nicolau 2020; Saldanha and Nardi 2016), including behavior analysis (Pinheiro and Mizael 2019, 2023). Brazilian feminist behavior-analytic studies are more recent and incipient than North American academic production, beginning a systematic dialogue with feminism in the 2010s (Couto and Dittrich 2017; Mizael 2021; Rosendo and Nogueira 2020). Some milestones of that endeavor are the creation of the Coletivo Feminista Marias & Amélias de Mulheres Analistas do Comportamento (the Marias & Amélias Feminist Group of Women Behavior Analysts), the formation of interest groups at meetings in the area, the holding of lectures and courses, and the production of articles, books, and academic works (Mizael 2021).

Furthermore, Brazilian behavior analysts have revitalized radical behaviorism (the philosophical dimension of behavior analysis) with crucial concepts for the feminist field. For example, as anti-essentialism is a philosophical commitment of radical behaviorism (Lopes et al. 2018; Villa and Melo 2021), some behavior-analytic studies have highlighted a non-essentialist conception of the female gender (e.g., Nicolodi and Hunziker 2021; Silva and Laurenti 2016). At the same time, the concept of subjectivity has also been discussed from a contextual and non-individualistic perspective (Lopes 2006; Pompermaier and Lopes 2018; Tourinho 2009).

However, it is still necessary to comprehend the subjectivation process of women from a radical behaviorist standpoint. Thus, this essay aims to outline a theory of subjectivity based on radical behaviorism and demonstrate its potential to discuss female gender aligned with feminist studies. I focus on four theses about subjectivity extracted from Brazilian behavior-analytic literature (subjectivity is a contextualized, multidimensional, pluralized, and politicized process). I apply them to understand women's subjectivation, considering, in particular, Brazilian feminist behavior-analytic studies. With a similar purpose to that of Mizael (2021), this delimitation intends to give visibility to a sample (even if small) of Brazilian academic production on subjectivity and female gender issues, hoping that new networks of researchers, whether behavior analysts or not, can be established.

The focus is on the female gender and does not encompass other gender possibilities, such as male, nonbinary, genderqueer, or genderfluid. In addition, much of the following discussion about women's subjectivation is also within a binary, patriarchal, and cis-heteronormative gender system due to its predominance in Westernized society, as in the Brazilian one, and not because it is considered the only possible and desirable system (Laurenti 2023; Nicolodi and Hunziker 2021; Pinheiro and Mizael 2019, 2023).

Despite the limitations of that analytical approach, I intend to demonstrate that behavior analysis (and its philosophy—radical behaviorism) can be an epistemological and political ally to emancipatory feminist studies. With that, I bring to the debate some efforts from the behavior-analytic domain to insert a feminist perspective into psychology—initiatives generally neglected in articulations between psychology and feminism and in endeavors to make explicit psychology's contributions to feminist studies (cf. Nuernberg et al. 2011; Rutherford 2012; Saldanha and Nardi 2016).

## 2. (Women's) Subjectivity in Radical Behaviorism

Considering some Brazilian behavior-analytic studies that have discussed the notion of subjectivity (Lopes et al. 2018; Pompermaier and Lopes 2018; Tourinho 2006a, 2006b), it is possible to delimit it through four theses: 1. Subjectivity is contextualized (rather than

internalized). 2. Subjectivity is multidimensional (embodied, personal, and reflective). 3. Subjectivity is plural (diverse, ambiguous, and conflicting). 4. Subjectivity is "politicized" (inscribed in power relations).

Below, I present the theses, applying them to the constitution of women's subjectivity—although the original behaviorist studies did not do so (including those of Skinner). Thus, I intend to outline an example of how this theory of subjectivity can approach feminist studies.

### 2.1. Contextualized Subjectivity

Sociologists have analyzed the historical, economic, philosophical, and political conditions of forming an internalized subjectivity, which gave rise to the emergence of modern psychology (e.g., Elias [1982] 1993; Sennett [1974] 1995). From this modern perspective, psychology would identify subjectivity with something internal to the individual, and therefore, interiority would define the psychological phenomenon. Psychological phenomena (e.g., thoughts, feelings, desires) would then be "occurrences of/in the individual" (Tourinho 2006b, p. 4).

Based on this modern view, women's subjectivity would be the same as their private experience. At the same time, knowledge of one's own psychological life would be hazy and confusing because, being subjective, it would be special, not exhibiting the required properties for objective and precise knowledge. Thus, a psychology that proposes to deeply understand women would be interested in scrutinizing their internal world, as it is from it that women make judgments about themselves and the world around them.

Radical behaviorism does not endorse that modern conception of subjectivity. Therefore, it does not equate subjectivity to interiority. So-called psychological phenomena are subjective, in a trivial sense, because they refer to a particular subject, and this does not mean they are inside that subject (Skinner [1971] 1974). For radical behaviorist philosophy, psychological phenomena are *behavior* (Skinner 1945; Tourinho 2006a, 2006b). *Behavior*, in turn, is the continuous relationship between the individual and the environment (social and non-social).

The first consequence of this premise is that psychological phenomena do not have an extra-mundane nature. If the individual is forged in a relationship with the world, the world integrates the human being even in its most particular aspects, such as the psychological one (Lopes et al. 2018).

Another consequence is that if psychological phenomena are behavior, and behavior is a relation, then psychological phenomena are relations (Pompermaier and Lopes 2018). Relations are neither internal nor external (Tourinho 2006b). Therefore, a behavior-analytic notion of subjectivity is neither internalist nor externalist but relational. *Subjectivity* describes particular or subjective relationships between the individual and the world, in which individuals are constituted as someone different from others—a difference perceived by others and felt and perceived by the individual. In this way, what is considered particular or subjective "are not internal and personal occurrences of an individual but their relationships with the world. They are not particularities of each individual but particularities of relationships" (Tourinho 2006b, p. 6).

The radical behaviorist understanding of women's subjectivation processes presupposes that relational perspective. Women's subjectivity will not be scrutinized by examining their inner life. Neither will it be anchored in a transcendent world populated by immutable archetypes or essences. Nor is it a question of stating that the genesis of subjectivity resides in social relations and then reinserting it as occurrences of/in women. The subjective phenomena produced in women's ties with *this* world continue to be relations. The "radical" attribute of behaviorism requires permanence in behavior to understand how women are constituted and interact as such.

*2.2. Multidimensional Subjectivity*

Radical behaviorism makes another withdrawal from the modern conception of subjectivity. In addition to rejecting the idea of interiority, this philosophy does not reduce subjectivity to the "self", whether in its rationalist ("self" as Reason) or romantic ("self" as feelings, desires, and emotions) version. Besides the "self", further dimensions of subjectivity are considered, such as the body and the person (Lopes et al. 2018; Skinner 1989).

The multidimensional conception of subjectivity acknowledges that body, person, and "self" are intertwined dimensions, so they cannot alone elucidate the complexity of women's subjectivation process. For example, the body cannot be thought of outside linguistic practices, as it is only within the scope of these practices that the body is part of the subjective dimension of *women*. Sensorimotor characteristics and body limits gain cultural meanings that do not necessarily correspond to the phylogenetic "reasons" forged in the evolution of species (see Skinner 1981, p. 503). As it is a body already born and developed in a given culture, it is impossible to separate this body from the social environment to understand women's subjectivity. The analytical starting point is a culture-laden body. Because of this, biological reductionist arguments, which invoke bodily characteristics (anatomic-physiological) to explain individual behaviors of women in a given culture, are inappropriate here.

On the other hand, the subjectivity field is not simply a disembodied reproduction of verbal cultural practices. Such practices come to life *with* (and not inside) a body, which can impose limits or indicate potentialities not glimpsed by those practices. So, cultural practices must be related to human bodies with a particular phylogenetic history. Cultural reductionism also finds no place here, as a multidimensional concept of subjectivity includes an embodied notion of culture.

The subjectivation process is reduced neither to the body nor to culture, as it is related to a singular individual. However, this individual is not the passive product of these dimensions. New destinies for both body and culture can emerge from the individual's personal history. The epistemic autonomy of the individual does not imply a psychological reductionism because, as already indicated, subjectivity is not defined by the personal dimension singly but by a culture-laden personal body.

Without losing sight of that emerging and dialectical dynamic, I divide the subjective dimensions to understand the particularities of each one better.

2.2.1. Embodied and Personal Subjectivity

In a primary sense, human relations with the world are already singular at birth. Even if human beings share characteristics with other members of their species, each one has a unique body from a biological point of view because no individual organism is the same as another, as Skinner ([1971] 1976) elucidated: "Every cell in his body is a unique genetic product, as unique as that classic mark of individuality, the fingerprint" (p. 204).

As a product of natural selection, such a body is a sensitive biochemical structure, being able to preserve or change its internal conditions depending on the exchange with the environment, maintaining organic balance (Skinner 1989). In addition to being sensitive, this body is active, as it was "prepared" to *react* (through reflex relationships, for example) to environmental changes favorable to its survival. In another sense, this sensitive body is active, as it can *act* (operant behaviors) in different ways in function with the effects of its activities in the environment (Lopes et al. 2018).

That sensitive and active body gradually becomes an organic unit of its own, in which its limits are progressively perceived. These limits change over time (body dimensions change with the age of the organism) quite slowly, establishing a sufficiently stable context for the constitution of a first sense of subjectivity: a body of one's own (Lopes et al. 2018; Skinner [1971] 1976).

As the body constitutes itself in relation to the environment, it acquires its own behavioral repertoire, which Skinner (1989) named "person": "Starting with the organism that has evolved through natural selection, they [contingencies of operant reinforcement]

build the behavioral repertoires called persons" (p. 28). This organic unit becomes a *person* throughout a unique life history; therefore, no one will behave the same way as others (see Skinner [1971] 1974, p. 168). The individual's uniqueness is already constituted as a personal body even before learning to speak. Still, it is not a question of saying now that this body *has* a person inside, but of "a body which *is* a person in the sense that it displays a complex repertoire of behavior" (Skinner [1971] 1976, p. 195).

The body-person unit is gradually formed and maintained concerning a primarily social environment. The variables that participate in the constitution and maintenance of the embodied personal behavioral repertoire are arranged by other people according to the cultural practices of the society in which they live (Skinner [1953] 2005). Individuals acquire a social identity when others recognize them as a singular body-person unit that is relatively stable and, at the same time, different from others. In other words, the individual is recognized by others as unique and "the same". Skinner ([1971] 1974) exemplified the point: "We refer to the fact that there is no one like him as a person when we speak of his identity. (The Latin *idem* means same, and when asked whether someone is really so-and-so, we may reply colloquially, 'The same!' or, 'Himself!'" (p. 168).

From a radical behaviorist perspective, the subjective relations that constitute "women" are embodied, involving a sensitive and active body. However, the bodily dimension is not enough to define women. There is also a personal dimension along with the body; no "person" is without a body, but the body only defines itself as its own with a personal behavioral repertoire. In an initial sense, "woman" is a social identity constituted from the recognition by others of a specific body-person unit (Silva and Laurenti 2016). One says, for example, that this *person* is a woman based on the observation of bodily characteristics (presence or absence of certain primary and secondary sexual characters) and behavioral characteristics (ways of speaking, sitting, walking, gesturing, dressing) integrated into a relatively stable bodily and behavioral whole. (As we will see later, this observation is not neutral but permeated by historically constituted social contingencies that "select" which aspects count to identify a person as a woman). The relative stability of the body-person unity relies not only on slow and gradual organic changes over time but also on stable physical and social conditions. For example, the members of a verbal community recognize that body person as the "same" woman, calling it by the same feminine name and determining spaces and social roles that only these body-persons designated as women can circulate and play (Silva and Laurenti 2016).

Up to this point, the subjective relations constituting the human being as unique involve a bodily and personal dimension. However, as already pointed out, this body-person unit is constituted and maintained in a verbal social environment, which is also responsible for the formation of another subjective dimension, capable of leading this body-person unit to perceive and feel as "itself" (the "self") (Skinner 1989). Therefore, "a woman" designates what others perceive (that person is a woman) and something felt and perceived as such: "I am a woman".

### 2.2.2. Reflexive Subjectivity

In social environments, the verbal community recognizes the subject as unique and creates conditions for individuals to identify themselves as singular. In other words, the social environment also constitutes the "self" or "self-knowledge" (Skinner 1957, [1971] 1974). The verbal community describes the body-person unit and asks individuals to describe their biological characteristics, bodily conditions, and behavior. It is within these verbal social contingencies that the notion of "self" emerges.

Through questioning such as "What did you do?" or "What are you doing?", people come to name the actions they emit as their own and to be controlled by this description. Through inquiries such as "What did you feel?" or "What are you feeling?", individuals learn not only to respond to certain stimulations produced by their organism but become able to verbally respond to their bodily conditions, "introspecting" them (Malacrida and

Laurenti 2018). What is introspected, therefore, is not an immaterial mind or even the brain but the body itself acting in the world (Skinner [1971] 1974).

Unlike the body-person unit, which others can observe, the "self" "is observed only through feeling or introspection" (Skinner 1989, p. 28). While others might even agree with the sentence "I was a different person", they can no longer do so when one says "I was not myself" because the statement indicates "that I felt like a different person. The self is what a person *feels like*" (p. 28).

Additionally, the verbal community arranges conditions for people to explain why they feel and behave in such a way. By answering questions like "Why did you do that?" or "Why did you feel that?", individuals are led to identify the "causes" of what they do and feel (see Skinner [1971] 1974, p. 141). In this sense, "self-knowledge is of social origin" (Skinner [1971] 1974, p. 169).

The teaching of self-description and self-knowledge repertoires is ruled by the society's value system of which the individual is part. Through ethical control, the verbal community teaches individuals to value their bodies and behaviors as good or bad, correct or incorrect, virtuous or sinful, normal or abnormal, worthy or unworthy, and useful or useless (Skinner [1953] 2005). For example, individuals come to value themselves positively (self-esteem) when "a culture commends and rewards those of its members who do useful or interesting things, in part by calling them and the things they do good or right" (Skinner 1989, p. 30). Being the behavior of these people positively reinforced, "bodily conditions are generated that may be observed and valued by the person whose self it is" (p. 30).

The verbal community requires that individuals not only know and value their behaviors and bodies but also change themselves according to society's ethical system. In this sense, a given verbal community can foster individuals who do not let themselves be carried away by emotion and make wise and non-impulsive choices (emotional self-control) (see Skinner [1953] 2005, chp. 15); individuals who "think before they speak", being able to edit their verbal behavior for themselves before making it public (self-editing) (see Skinner 1957, chp. 15, p. 371); and autonomous individuals, being able to think and solve problems for themselves (intellectual self-government) (see Skinner 1968, chps. 6 and 8).

All these so-called psychological activities, commonly named self-determination or choice (self-control), thinking, and creativity (self-editing and intellectual self-government), are behavioral relations (see Skinner [1953] 2005, p. 228). In many of these activities, the individual manipulates variables that are not always accessible to others. Still, it does not mean they are internal psychological activities (Skinner [1953] 2005). Even the "self", one of the psychological phenomena par excellence, has also been re-described in behavioral terms: "The 'self' is a product of verbal cultural practices that shape and maintain a first-person narrative" (Lopes et al. 2018, p. 82). There is, therefore, no "self" that pre-exists behavior or a "self" that is described by verbal behavior because the "self" *is* verbal behavior (Malacrida and Laurenti 2018).

In societies based on a binary gender system as a form of social organization, the possibility of a personal body designating itself as a woman ("I am a woman") is grounded in the gendered verbal practices of the verbal community of which it is part. In such binary practices, social contingencies are in place that link (male/female) gender designations to the teaching of reflexive repertoires. It is a matter of teaching such repertoires and consistently reinforcing the gendered emission of these self-reports. The person begins to see and feel as a woman ("I feel like a woman") (self-observation); to describe herself as a woman ("I am a woman") (self-description); to speak to herself as a woman ("I think of myself as a woman") (self-editing); to choose (self-control) and solve problems (self-government) as a woman; and to value herself as one according to the ethical system of verbal communities ("I am a good/bad, normal/abnormal, virtuous/sinful woman").

The *behavioral* nature of the first-person gender perspective shows that the emission of the verbal response "I am a woman" is relational and inherently contextual. Its *social* nature highlights that these self-reports are constituted in contingencies organized by other people. The *verbal* nature of gendered self-reports reminds us that the verbal response "I am a

woman" will only continue to be maintained through reinforcement provided by listeners, trained by a verbal community following the gendered cultural practices of their culture.

Given these premises, perceiving, feeling, and defining herself as a woman is not an expression of a feminine essence but is the actualization in a personal body of verbal practices that make up a system of social reinforcement of gendered self-reports. This system "sustains" the force of emission of these verbal responses, not essences. It is, therefore, necessary to scrutinize the social history of those gendered verbal practices, and not to appeal to timeless models or ideal types, to understand why certain personal bodies perceive and feel like women and why they often do so similarly to other women (i.e., the aspects used to define themselves as women are culturally shared).

*2.3. Pluralized Subjectivity*

In addition to being multidimensional, subjectivity is also pluralized. It means that each subjective dimension (bodily, personal, and reflexive) changes over time, diversifies, and interrelates, not always cohesively and harmoniously (Lopes et al. 2018). Although relatively stable, the body changes due to age, diseases, and cultural practices (e.g., diets, surgeries, physical exercises, medications, and other substances). The personal dimension is plural, too; the contingencies of reinforcement that constitute the person are diverse, so different social contexts form different people or repertoires coexisting in the same body (see Skinner 1989, p. 28). Similarly, self-reflexive repertoires are generated and maintained by a social environment that is plural likewise, constituted by heterogeneous verbal communities (Skinner [1971] 1974, 1989).

The different and diverse subjective dimensions intertwine and change in this intertwining. Although verbal, the "self" is embodied in a body because behavior belongs to the body. Thus, when a first-person verbal operant is actualized, this actualization takes place *with* the body, involving emotions and feelings. The verbal social contingencies constitutive of the "self" are also based on observing individuals' behavior patterns in different contexts (i.e., persons), many of which individuals are unconscious of—even because "without the help of a verbal community all behavior would be unconscious" (Skinner [1971] 1976, p. 187).

The "self", in turn, is not merely a description or "grasping" of the personal body; it redefines this unit according to the social contingencies organized by verbal communities. The bodily dimension, for example, is influenced by the first-person description and not merely by physical limits—it is no longer a matter of distinguishing the body from the world but of talking about the body and attributing its qualities or flaws (Lopes et al. 2018). Such re-descriptions, therefore, affect the body and the person, which, in turn, likewise affect the "self", composing a back-fed circuit mediated by the social context.

The subjective dimensions of the individual can remain in relative harmony as long as the contexts remain stable and relatively independent, but this is hardly the case (see Skinner 1990, p. 1207). The social environment is much more unstable, variable, and unpredictable than the non-social one and, therefore, quite prone to generating conflicts (Skinner [1953] 2005). So, the individual is constituted among plural subjective dimensions (bodily, personal, and reflexive) whose relations can be ambiguous and conflicting. Again, these conflicts do not occur *inside* but *with* the individual's body in different relationships with plural contexts.

The plural (i.e., relatively diverse and incongruous) aspect of subjectivity implies that the female gender is not a static, self-enclosed, harmonious condition. The subjective dimensions that constitute women, although relatively stable, are changeable and conflicting. The possibilities for change and the measure of conflicts depend on the society where these body-person-self units called women live. For example, in a patriarchal and cis-heteronormative society, female gender possibilities that dissent from those imposed by the dominant gender system (i.e., woman with vagina, feminine, and cis-heterosexual) are often punished. Women who display bodies, behave, and feel differently from the dominant gender system are constantly exposed to intense conflict. The alternative contingencies that permeate the subjective dimensions of these women collide with the dominant

gender contingencies that control crucial groups in our society. Thus, the relative stability of the (only) possibility of the female gender retained by the dominant system is also due to oppressive practices.

### 2.4. 'Politicized' Subjectivities

If the relationships that single out bodies as reflexive persons are established and maintained by a verbal social environment, then they are, in principle, permeated by power relationships. Therefore, the discussion of subjectivity necessarily involves a political dimension. The social contingencies are organized by more or less powerful groups and individuals in a given society. Some forms of social control give certain groups and individuals more conditions to organize social contingencies that affect others than the opposite (Skinner [1953] 2005).

In Westernized societies centered on institutional control, "controlling agencies", such as government, religion, psychotherapy, economy, and education, are examples of powerful groups. Their representatives (the controllers) have more power to dispose of the social contingencies that shape and maintain the behavior of the controlees (Skinner [1953] 2005).

These institutions have a particular interest in regulating the social contingencies that forge subjectivities, especially those contingencies responsible for the "self" (Skinner [1953] 2005). Controlling agencies aim, above all, to establish repertoires of self-control and intellectual self-government that will enable individuals to behave according to institutional practices on different occasions and in the absence of any agency representative (e.g., priest, police officer, psychologist, boss, teacher) (Skinner [1953] 2005). In short, the constitution of the "self" by the institutions is also a form of social control[1] (Malacrida and Laurenti 2018).

For that reason, controlling agencies organize verbal social contingencies that favor the formation of obedient individuals. Skinner ([1953] 2005) explained the point: "By establishing obedient behavior, the controlling agency prepares for future occasions which it cannot otherwise foresee and for which an explicit repertoire cannot, therefore, be prepared in advance" (p. 338). Thus, "when novel occasions arise to which the individual possesses no response, he simply does as he is told" (p. 338).

The agencies' justification is that obedient individuals ultimately promote the common good. Due to their broad power to control, in principle, only institutions could guarantee collective goods such as justice, security, freedom, salvation, mental health, and knowledge (Skinner [1953] 2005). If this is so, an obedient individual, by acting in conformity with the controlling practices of agencies, would, by extension, be securing these goods.

Agencies, however, often use the power of control for their own benefit: "Religious agencies, like all other agencies here being considered, have sometimes used their power for personal or institutional advantages—to build organizations, to accumulate wealth, to punish those who do not come under control easily, and so on" (Skinner [1953] 2005, p. 358). Furthermore, they use various controlling techniques to restrain the possibilities of counter-control by the controlee—because it is precisely through counter-control that the organized agencies' power can be restricted (Skinner [1953] 2005, [1971] 1976). If so, the constitution of obedient individuals would be another institutional control strategy employed to hinder counter-control by controlee. Using reflexive repertoires to adjust to the practices of agencies, an obedient individual would be acting not in favor of the common good at all but for maintaining such institutions' unequal system of power and privilege.

Brazilian feminist behavior-analytic studies have expanded Skinnerian analyses, showing that in patriarchal Westernized societies permeated by racism and cis-heteronormativity, White men are the leading representatives of controlling agencies. For example, women are still a minority in the Brazilian National Congress, with the number of Black and transgender women being even more diminutive (Fontana and Laurenti 2020). Disparities like these also occur in the most prestigious activities in science in general and behavior analysis in particular (Freitas and de Morais 2019; Laurenti et al. 2019; Nicolodi and Hunziker 2021; Teixeira Silva and Arantes 2019).

In those social environments, the gender hierarchy permeates the constitutive contingencies of the multiple dimensions of subjectivity (body, person, and "self"). The verbal community not only names certain personal bodies as "women" but also negatively values them. Thus, biological and behavioral characteristics named feminine are downgraded, compared to those said to be masculine, which are exalted. This female inferiorization is accentuated with greater degrees of dehumanization when it comes to non-White and non-cis-heterosexual women (Fontana and Laurenti 2020; Laurenti and Lopes 2022; Mizael et al. 2023).

Patriarchal controlling agencies contribute to establishing a gendered system of obedience, in which women must be submissive to men. Even though they have different powers and ethical systems, masculinist controlling agencies cooperate in perpetuating cultural practices of male domination, ensuring the ubiquity of these practices in society. In sum, controlling agencies institutionalize male domination, establishing the patriarchy (Nicolodi and Hunziker 2021). Psychotherapy is not free from the reproduction of these practices and may, inadvertently or not, strengthen patriarchal contingencies of masculinity and femininity associated with other contingencies of domination and exploitation (Backschat and Laurenti 2023; Costa 2019; Pinheiro and Oshiro 2019).

### 2.4.1. On the Construction of Women's Submission

The dissemination by the controlling agencies of cultural practices shaping "feminine" as "submissive to men" favors the system of male privilege, which gives men greater possibilities to establish contingencies for women than vice versa. In gender oppressive contingencies, "being a woman" means maintaining, consciously or not, power asymmetries in favor of men. Masculinist institutions use different social control techniques to maintain male supremacy, some of which are violence and the concealment of oppression.

### Violence

Submission is a product of oppression, that is, of systematic practices of domination-exploitation of women (Nicolodi and Hunziker 2021). For example, the establishment of obedient women involves violence. Practices of violence against women feed back into the imbalance of power and the system of privileges enjoyed by men in a patriarchal society (Nicolodi and Hunziker 2021). From a behaviorist perspective, violent behaviors directed against women are not isolated and idiosyncratic actions of intrinsically aggressive individuals but part of a "rape culture"—a social environment that encourages, tolerates, trivializes, and naturalizes sexual violence by men against women (Freitas and de Morais 2019).

Women's bodies are also objectified, i.e., treated as things that can be consumed and discarded (Fontana and Laurenti 2020). In Brazil, Black bodies are even more sexualized than White ones, with Black women considered more prone to debauchery due to their complexion (Mizael et al. 2023). Moreover, Brazil still leads the global ranking of LGBTQIA+ murders for the 14th consecutive year. At the same time, it is the country that most watches trans pornography (Benevides 2023), with Brazilians being more likely to consume pornography featuring trans women compared to the rest of the world (Dias et al. 2022).

Forms of violent social control that are more difficult to recognize and describe ("symbolic violence") also contribute to women's obedience. The division of spaces and activities between men and women, the inferiorization of femininity, the conformation of patriarchal aesthetic standards, and female sexual practices aimed at male fulfillment are some examples of these practices (see Fontana and Laurenti 2020).

All of them produce pernicious effects on women's lives. At the psychological level, for example, the inferiorization of femininity generates low self-confidence and low self-esteem. As the "self" emerges from verbal social contingencies, women may perceive themselves as inferior to men in the patriarchal order of gender. Moreover, in the case of Black women, in particular, the intersection between race and gender contributes to even more acute experiences of devaluation, resulting in lower self-esteem compared to White women and feelings of insignificance and emotional neglect, as described by the "Black women's loneliness" concept (Mizael et al. 2023). Studies indicate that non-cis-heterosexual



women have to deal with stressors associated with a hostile environment of intolerance, prejudice, discrimination, and violence, which are predictors of mental health issues, such as depression and anxiety (Laurenti and Lopes 2022).

Concealment of Gender-Oppressive Controls

In societies marked by gender oppression, controlling agencies also disseminate verbal practices that conceal or distort the sources of control, making it difficult for the controlee to counter-control. Skinner ([1971] 1976, [1971] 1974) persistently criticized three of them: mentalism, the classical liberal conception of the individual as a free agent and initiator, and freedom as feeling. All these verbal practices constitute fertile ground for the depoliticization of the feminist debate in psychology.

*The mind as the cause of behavior*

Mentalism consists of verbal social contingencies that reinforce attributions of internal causes to individuals (Skinner [1971] 1974). The notion that the mind is the cause of behavior or that subjectivity is interiority are paradigmatic examples of this verbal practice. From a political point of view, mentalistic verbal practices shift the lens away from social contingencies, blaming the individual for their success or failure in society (Laurenti and Lopes 2022).

For example, patriarchal mentalistic explanations attribute specific inner causes to women, such as emotion, sensitivity, insecurity, irrationality, and tenderness, that serve as a social justification for placing them in less privileged roles and positions in the patriarchal hierarchy (e.g., in reproductive work and the domestic sphere).

At the same time, mentalistic verbal practices have infiltrated the feminist lexicon, leading to a depoliticization of feminist issues. For example, empowerment and female agency have been used to describe an "internal quality of the individual" (Rutherford 2018, p. 623) or a "psychologized feeling" (Rutherford 2018, p. 624). The psychologized and depoliticized use of feminist terms lead women to look inside themselves to understand gender oppression, not at historical and social contingencies (Lopez 2023).

*Humans as free and autonomous agents*

The notion of a rational, autonomous, and free individual anchored in classical liberalism is another verbal practice objected to by radical behaviorism (Skinner [1971] 1976). For that philosophy, the individual is not autonomous and free if autonomy and freedom are decontextualized activities. It is worth remembering that the rational and agentic qualities that would characterize the individual as "free" are socially constructed verbal repertoires because it is "a social environment that contains the contingencies generating self-knowledge and self-control" (Skinner 1978, p. 52). From a political point of view, the understanding that these human attributes are formed apart from the social contexts in which individuals live ends up hiding the unequal distribution of social contingencies responsible for developing these attributes (Laurenti and Lopes 2022).

"Liberal" patriarchal discourse is seductive because it portrays women as free agents rather than as submissive to men. The woman is "free" because she can rise socially and do what she wants with her body, desire, and sexuality. Ultimately, a free (rational and autonomous) woman is a successful woman (Rutherford 2018).

The problem with those verbal practices is that they camouflage the unequal gender distribution of the social contingencies responsible for making "rational" and "free" individuals. In this way, gender-oppressive contingencies will hardly provide conditions for women to be successful, except for some already privileged ones. Moreover, the failure to be entirely "free" would be the responsibility of the women, who did not try hard enough, and not of the gender asymmetries of the patriarchal society. This type of verbal practice also distorts an emancipatory feminist project of women's liberation, which seeks social recognition of the category of women and substantive changes in patriarchal, racist, cis-heteronormative, and capitalist social structures (Lopez et al. 2023; Hooks [2000] 2015).

Additionally, the classical liberal notion of humanity is not equitable for all bodies, persons and selves (Laurenti 2023). In patriarchy, "humankind" is synonymous with "mankind". Women are considered less human than men because women would be intrinsically more emotional and less rational, more heteronomous than autonomous. Thus, their complexion would explain the condition of "less freedom" for women, which would justify the need to be tutored, monitored, curtailed, and dominated by men. Patriarchal dehumanization allied to racism places Black and Indigenous women further away from the liberal notion of the subject, as they would be closer to nature and, therefore, to emotionality and irrationality (Laurenti 2023). Patriarchal dehumanization coordinated with capitalism assigns women to reproductive work compensated only with the "currency" of love and affection, making them financially dependent on men. In the case of Black women, reproductive work adds to the burden of poorly paid salaried work, being more exploited than White women compared to men's earnings (Fontana and Laurenti 2020; Laurenti 2023).

In oppressive gender contingencies, dehumanizing women begins at birth and continues throughout life. The designation by others of a personal body as a "woman" seals the fate of this individual, even before self-designating as such: she is going to be a mother. In addition to heteronormative practices, her sexual desire is already directed: she is going to love men. Along with transphobic practices, her gender identity is also defined: she is going to be a cis woman. Her interest is predetermined in advance based on sexist practices: she is going to "like" care activities. Immersed in male chauvinism practices, she is never going to threaten men's protagonism. Permeated by misogynist practices, her body is objectified, her person disqualified, and her "self" despised. Added to classist and racist practices, if she is born rich, she is going to take care of the home; if poor, she is going to serve in others' homes; if White, her body is going to serve to bear children; if Black, to take care of the children of White women; if they are White, blonde and thin, they are going to be chosen, if Black or fat they are going to be deprived. Therefore, the condition of women in patriarchal, racist, and cis-heteronormative societies is not one of freedom.

*Freedom as feeling*

Another favorable verbal practice for concealing gender-oppressive controls is to describe freedom as a feeling: to be free is to feel free to do what one wants (Skinner [1971] 1976). From a political point of view, similar to mentalism, the struggle for freedom (the last one understood as a feeling) has concealed forms of unequal social control that do not rely on explicitly aversive techniques but positive reinforcement. For this reason, "the feeling of freedom becomes an unreliable guide to action as soon as would-be controllers turn to non-aversive measures, as they are likely to do to avoid the problems raised when the controllee escapes or attacks" (Skinner [1971] 1976, p. 37).

As anticipated, some notions of female empowerment have endorsed freedom as feelings of belonging, a strong sense of self and agency, self-confidence, self-esteem, and personal empowerment (Couto 2019). An "empowered" woman would feel free to do whatever she wants with her body (e.g., plastic surgery), with her desire (e.g., masturbation), and with her interests (e.g., consuming certain products, choosing certain professions), understanding that this "wanting" is constituted apart from her relations with the social context.

However, as already suggested, the feeling of freedom is not a good indicator that women are free (Couto 2019). Using non-aversive control techniques can produce the feeling of freedom as one of its effects. Thus, a woman may feel free and not rebel (i.e., not exercise counter-control), even though she is under unequal social control, which favors masculinist controlling agencies more than the woman herself in the long run. Skinner ([1971] 1976) named this condition the "happy slave": "The word 'slave' clarifies the nature of the ultimate consequences being considered: they are exploitative and hence aversive" and "a system of slavery so well designed that it does not breed revolt is the real threat" (p. 44). Describing oneself as empowered based only on what one feels does not a priori rule out the condition of the woman being a "happy slave".

2.4.2. Is It Possible for Women to Rise Up against Gender Oppression?

From a radical behaviorist perspective, women's obedience and submission are not explained by essential characteristics (biological, psychic, or metaphysical). Women are obedient or submissive because of oppressive social gender contingencies composed of different axes of domination-exploitation (patriarchy, racism, capitalism, cis-heteronormativity), whose intertwining makes women's subjectivation also a form of subjugation (Lopez et al. 2023). Controlling agencies reproduce these oppressive contingencies with potent controlling techniques, some more and others less explicit, making such contingencies ubiquitous and challenging to counter-control. If an intrinsic feminine nature does not trace the destiny of women, would it be by a social determinism?

Against this backdrop, the theoretical challenge of radical behaviorism is to demonstrate the possibility of women, even though they have been subjectively shaped in an oppressive social context that encourages obedience, to rise against that same context. Another challenge is to indicate the viability of this counter-control without underhandedly reintroducing the initiating "self" with a decontextualized notion of female agency to justify the possibility of women being free from oppression (Silva 2021).

In the ontological sense, the behavior assures the possibility of insurgency. The individual is forged in behavior, and behavior is a processual, "changing", "fluid", and "evanescent" relation (Skinner [1953] 2005, p. 15). Although regularities can be established within this individual–environment relationship, they are open to change[2]. Skinner ([1971] 1976) also emphasized that individuals are simultaneously controlling and controlled; they modify the environment and are not only unilaterally modified by it: "Men [sic] act upon the world, and change it, and are changed in turn by the consequences of their action" (Skinner 1957, p. 1). In sum, behavior comprises dialectical relationships between the individual and the environment, in which the individual constitutes and is constituted by the environment. If so, women can virtually do things differently from the fate envisioned by oppressive gender contingencies.

This ontological possibility cannot be held lightly, placing the responsibility for doing differently on women and reintroducing a decontextualized notion of female agency. Concrete conditions are necessary to actualize the ontological possibility of doing differently from oppressive gender contingencies. Some of them are supported by pluralism (Lopes et al. 2018). As we have seen, a social environment is plural, more inconsistent than the non-social environment, and does not affect individuals similarly, so "it is also probably never the same for two individuals" (Skinner [1953] 2005, p. 424). Even in the case of a social environment organized by controlling agencies, there is heterogeneity, as some are more powerful than others (see Skinner [1953] 2005, p. 424). Thus, the concrete possibility of acting in alternative ways to what is established by oppressive gender contingencies arises on the margins of plural social contexts.

The political task of psychology is to explore the gaps in these oppressive social controls, contesting their mentalistic verbal practices and composing alternative verbal communities for an emancipatory process of women's subjectivation (see Lopez et al. 2023).

First, these alternative verbal communities must *expose the oppressive controls* that permeate women's subjectivation processes (Lopez et al. 2023; Silva 2021). In addition, the self-knowledge provided by these communities should foster self-reports that connect descriptions of what women do, think, and feel with contextual variables. The emphasis on women's relationships with the social context does not imply ignoring feelings because knowledge is embodied. As mentioned, feelings must be contextualized and not considered the compass of reality alone. The focus on a contextualized feeling does not fail to nurture feelings of self-confidence and self-esteem. Quite the contrary, because in welcoming social environments that promote positive feelings, people tend to adhere to and perpetuate their practices (Skinner [1971] 1976). The point is that these feelings are not the effects of verbal practices of individualistic self-aggrandizement and do not camouflage oppressive controls (Skinner [1971] 1976, 1978).

While meaningful, the Skinnerian proposal for explicit social controls can be enhanced by adding analytical tools from Black feminism, such as intersectionality (Lopez et al. 2023; Mizael 2019). An intersectional understanding of oppression highlights the limits of binary and static categories, such as controllers (men) versus controlees (women). Indeed, women have less power of control than men in a patriarchal society. However, this asymmetry of power occurs in different ways considering the "knot" or "entanglement" of patriarchy with other axes of domination-exploitation (e.g., capitalism, racism, colonialism) (Lopez 2023; Nicolodi and Hunziker 2021). In this way, even though men oppress women, women can also oppress other women, such as rich women oppressing poor women, White women oppressing Black and Indigenous ones, and women from the Global North oppressing those from the Global South. This complex scenario makes the struggle against oppression a plural and collective struggle by and for all women (Lopez et al. 2023).

In the second place, these disruptive verbal communities need to *prepare women for effective counter-control*, as it is one way to limit controllers' power. Strengthening counter-control is even more critical when gendered verbal practices blur oppressive control (Skinner [1971] 1976). One strategy is to create verbal contexts in which marginalized groups can formulate descriptions of themselves and the world through alternative verbal practices to those of dominant groups. Such re-descriptions are crucial to changing the symbolic controls that render these groups unworthy of recognition by dominant verbal practices. They are also necessary for marginalized groups in devising their struggle strategies to radically transform social structures toward more egalitarian and just conditions (Lopez 2023).

The last point is that *members of these disruptive verbal communities cannot be committed to the social control employed by controlling agencies* (Skinner 1987). The ends do not justify the means: one cannot draw upon hierarchical and unjust controls to promote more egalitarian and fair controls. For Skinner (1987), "uncommitted" individuals "to governments, religions, and capital" are "free to consider a more remote future" but will be truly free "only to the extent that we are, in fact, uncommitted. If there are leaders among us in government, religion, and business, they are with us only insofar as they are uncommitted to their respective institutions" (p. 8).

Despite criticism of controlling agencies, Skinner (1978) recognized the difficulty for societies like ours to dispense with institutional control as a structure of social organization. Therefore, it is necessary to widen the fissures in this type of control and inoculate other forms of social relations (e.g., face-to-face control) that do not reproduce such power imbalances (see Skinner 1978). It is also necessary to reinvigorate counter-control strategies to restrict the abusive power of agencies and make institutions more sensitive to the consequences of their control practices (Skinner [1953] 2005).

Skinner (1989) believed that scientists, artists, and intellectuals could contribute to outlining alternative forms of social control. In theory, these fields of knowledge constitute verbal communities that share forms of social control that would make their members more sensitive to social misfortunes and the future of humanity. Hence, the Skinnerian argument that specialized knowledge spreads beyond its limits and advances in the dispute for common sense. The more scientific verbal practices reach the controlee, the less likely they are to be co-opted by agencies, as Skinner ([1953] 2005) argued: "By distributing scientific knowledge as widely as possible, we gain some assurance that it will not be impounded by any one agency for its own aggrandizement" (p. 442). However, scientific, artistic, and academic verbal communities have also been porous to oppressive gender control, either by working in their favor or becoming one (Laurenti et al. 2019; Teixeira Silva and Arantes 2019).

The possibility of establishing alternative forms of social control to the institutional one is revitalized with the revolutionary vein of Black feminism, which claims the participation of marginalized groups in constructing these controls (Lopez 2023). So, the strategies suggested by Skinner can be combined with those of Black feminism. "Uncommitted" members of the center (e.g., scientists, artists, scholars) can articulate with groups on the margins and, together, build possibilities of social control capable of, hopefully, imploding the unequal contingencies that generate the very division between center and margin.

Indeed, the marginal location of these groups cannot automatically generate counter-controls and alternative formulations to hierarchical and unfair controls (Lopez 2023). Nevertheless, the fact that marginalized groups are exposed to and use verbal practices different from those employed by dominant groups makes them powerful contexts for envisioning more egalitarian and just forms of social control (Lopez 2023).

Within disruptive verbal communities, women can actualize the emancipatory onto-logical potentialities of behavior and constitute themselves as subversive subjectivities to oppressive gender control. Once established and maintained, these subjectivities strengthen the environments that nurture disruption. Thus, women's freedom can be achieved in this virtuous circuit between disruptive verbal communities and subversive subjectivities.

For radical behaviorism, "freedom is a matter of contingencies of reinforcement, not of the feelings the contingencies generate. The distinction is particularly important when the contingencies do not generate escape or counterattack" (Skinner [1971] 1976, p. 42). From this angle, the struggle for women's liberation would not be a struggle for the absence of control, mainly because, for that philosophy, control is inevitable. Women's liberation is the struggle for the constitution of emancipatory social environments, for forms of social control in which women can actualize the multiple dimensions of subjectivity in balanced and fair relations of control and counter-control.

In sum, a woman is free when her self-descriptions and the changes she makes to her body and her environment are not in the service of forms of social control that benefit men at the expense of women, nor are they condescending to power inequalities among women.

## 3. Final Remarks

Radical behaviorism is an anti-essentialist philosophy (Lopes et al. 2018; Silva and Laurenti 2016). This non-essentialist perspective is an interface area with emancipatory feminist projects. This essay aimed to demonstrate that radical behaviorism also guides a notion of subjectivity capable of expanding the area of contact with feminist studies.

A contextualized, multidimensional, pluralized, and politicized theory of subjectivity allowed us to understand the construction of women's subjectivity without transcending the relationships between individuals and the social context. So, women's subjectivity is comprehended as particularities of relationships and not particularities of an inner experience.

Women are forged in a tripartite complex of body, person, and "self", whose relative unity is susceptible to change and conflict. In a patriarchal, racist, and cis-heteronormative society, such as the Brazilian one, subjectivation is also an oppressive process. Society restricts female gender possibilities to just one, which is based on the subjugation, exploitation, and inferiorization of women. Powerful institutions use violence and practices of hiding oppression to make other female gender possibilities unfeasible and strengthen this non-inclusive gender order.

On the other hand, women's subjectivation can be glimpsed as an emancipatory liberation process based on the relational and contextual assumptions of radical behaviorist theory of subjectivity. Without falling into some theoretical traps of feminism, such as the antinomy of social determinism versus agency (Ruiz 1998; Silva 2021), that psychological orientation indicates the possibilities of emancipation in dialectical circuits between disruptive verbal communities and subversive subjectivities.

Future works could improve the reflections evoked by this study. For example, it is necessary to expand sources beyond the Brazilian behavior-analytic literature prioritized here. At first glance, the four theses on subjectivity could be used to understand other possibilities of gender and subjectivity not explored. However, the theoretical potential and limitations of this proposal still need to be assessed in a broader interpretative endeavor. For example, the assertions on subjectivity could be enriched with a discussion on intersex bodies, nonbinary gender, and people with disabilities. Other axes of domination-exploitation, such as colonialism, and other social markers, such as age, religion, and class, must also be considered in further studies.

Even though it is a preliminary hermeneutic effort, this essay signals that the behavior-analytic field can situate the process of women's subjectivation not as a locus of depoliticization but as a pivotal ally in constructing a more equitable and just society as envisioned by feminism.

**Funding:** Carolina Laurenti was supported by the National Council for Scientific and Technological Development (CNPq) with a Research Productivity Grant (grant number: 315116/2021-8).

**Institutional Review Board Statement:** Not applicable.

**Informed Consent Statement:** Not applicable.

**Data Availability Statement:** Not applicable as no new data were created for this conceptual paper.

**Conflicts of Interest:** The author declares no conflict of interest.

## Notes

[1] Skinner does not use the term control in a pejorative sense. The assertion that the teaching of the self is a form of social control does not necessarily mean that it is a tool for dominating and exploiting personal bodies. The term control is undoubtedly indigestible, but Skinner's maintenance of its use highlights the inherently contextual condition of the human being. As we will see later, he attacks the decontextualized conception of the individual of classical liberalism, for which individuals are rational, free, and autonomous, regardless of their relations with the context (Skinner [1971] 1976). The environment controls human beings because they are continuously constituted and influenced by it. There are coercive controls, in which social control is intentionally exercised as a form of domination and exploitation. There are also, without contradiction, emancipatory controls in the sense that the mutual relations of control between individual and social context are balanced, edifying, and constructive (Skinner [1971] 1976).

[2] Opposing the notion of a decontextualized autonomous individual, Skinner argued that his system is deterministic and that the individual's activities are determined by the environment (e.g., Skinner 1968, p. 167; [1971] 1974, p. 54; [1971] 1976, p. 26; [1953] 2005, p. 6). Nevertheless, looking more closely, Skinnerian formulations of determinism depart from the traditional notion of the concept, based on which the individual's activities would be an inexorable and invariable effect of environmental causes, with no room for change. Moving away from this classical meaning, Skinner stated that the environment influences or controls the activities of organisms in a probabilistic way (see Skinner [1953] 2005, pp. 90–111), in addition to having recognized the role of chance in the relations between organism and environment (Skinner 1968, p. 180; [1971] 1974, p. 114; 1990, p. 1208).

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
