# Peer review of "Contributions of a “Brazilianized” Radical Behaviorist Theory of Subjectivity to the Feminist Debate on Women"

_socsci, doi:10.3390/socsci12110641_

Round 1

Reviewer 1 Report

Comments and Suggestions for Authors

ü  Line 87: Include, in parentheses, the four assertions or to mention that the four assertions will be described below

ü  Line 122: Even though Tourinho, as a radical behaviorist, makes this statement, I strongly suggest that Skinner's (1945) reference be included, since he addresses subjectivity as behavior.

Part of the relationships and reflections are unprecedented, with emphasis on the interpretation according to which subjectivity has a political dimension considering that, like other behavioral relationships, it involves power relations.

Despite being strongly grounded in Skinnerian literature, the article goes beyond this author's analyses, strongly reaffirming the potential for change of a radical behaviorist view.

Author Response

Dear Reviewer,

First of all, thank you for the suggestions. I sought to incorporate all of them in the revised version of the manuscript.

All changes are highlighted in red in the reworked manuscript file and described below.

Best regards,

Reviewer: Line 87: Include, in parentheses, the four assertions or to mention that the four assertions will be described below

Reply: I have included the four assertions in lines 76 and 77. See: I focus on four theses about subjectivity extracted from Brazilian behavior-analytic literature (subjectivity is a contextualized, multidimensional, pluralized, and politicized process).

Reviewer: Line 122 - Even though Tourinho, as a radical behaviorist, makes this statement, I strongly suggest that Skinner's (1945) reference be included, since he addresses subjectivity as behavior.

Reply: I have inserted the suggested reference of Skinner in line 127. See: For the radical behaviorist philosophy, psychological phenomena are behavior (Skinner 1945; Tourinho 2006a, 2006b).

Reviewer 2 Report

Comments and Suggestions for Authors

The paper proposes a thought-provoking discussion on the possibilities of radical behaviourism's contribution to feminist studies, analysing how this field of psychology defines subjectivity, the body and freedom, and how these notions dialogue with certain fields of feminism, especially those that do not intend to depoliticise feminist claims.

Since the work brings together heterogeneous theoretical fields and references, some points and arguments need to be better developed to make it easier to read and understand for audiences who are not specialists in radical behaviourism or feminist studies.

Here are some suggestions for improving the text - mostly minor revisions listed below. Two main aspects should be considered more generally in the text review:

1. make explicit the notion of gender that underpins the discussion: the text clearly defines the behavioural framework used, and dialogues this framework with discussions from feminist fields. However, it does not state which notion of gender underlies its discussion (there are different conceptualizations of gender produced by different feminist strands). At different points in the text, gender seems to equate to "women", which is problematic. At some points, even, to "cisgender women". The concept and its discussion must be organized in a more explicit and referenced way, from the beginning of the text;

2. review the choice of the concept of patriarchy: the paper uses the concept of patriarchy to explain gender inequalities and sexism. However, this concept is not the most accurate to sustain the debate, since it conflicts with the theories and reflections proposed by black and intersectional feminisms. The idea of patriarchy assumes that gender oppression is the core of inequalities, while black feminists and intersectional feminists point out that sexism is one of the axes of oppression, and racism must be articulated with sexism, class prejudice, LGBTQIAphobia, and other oppressions. The use of these perspectives conflicts with the centrality of the concept of patriarchy in the paper, which must be reviewed.

- The title of the article should be reformulated to address the rapprochement and contribution between the fields (radical behaviourism and feminist studies). The political meaning of the terms is discussed in the text, but does not seem to be the central issue of the work;

- The abstract should be revised. Up to line 12, it gives a generic justification of the article's problem, and the abstract ends up not including important definitions addressed in the work, such as the four theses developed on subjectivity;

- In the Introduction, on page 56, it is stated that "the objective is to explain the emancipatory potential of psychology from the perspective of Behavior Analysis.". However, this does not seem to be exactly the general objective of the paper. This passage should be revised;

- On page 4, we read "Product of natural selection, such a body is a sensitive biochemical structure, being able to preserve or change its internal conditions depending on the exchange with the environment, maintaining organic balance (Skinner 1989).". It would be important to point out that there are biased approaches to behavioural psychology that use the evolutionary framework to explain and justify social practices, behaviours and human phenomena crossed by gender inequality. This sexist use of evolutionary theories is not addressed in the paper, and it would be important to point it out, especially as the author refers to natural selection in explaining what a body is for radical behaviourism;

- Due to the lack of a definition of the concept of gender on which the article's discussion is based, at different points in the text it is not clear how the author understands the place of trans identities, especially trans women, in feminism. On page 8, in particular, the term identify (referring to trans people) is enclosed in quotation marks and the trans prefix is in italics. Why?

- Two passages, on page 7 (lines 315-317) and page 10 (lines 457-462), must be reviewed, in which it is not clear whether Skinner's statements refer to gender issues or individuals/concepts in general;

- On page 10 there is a critique of depoliticizing approaches to the concept of "lugar de fala", which is only defined later in the text. It is necessary to define the concept here, to then criticize neoliberal interpretations of it;

- The terms "aversive techniques", "positive reinforcement" and “non-aversive control techniques” are behaviourist concepts that must be explained (maybe in a footnote) for non-specialist readers.

Comments on the Quality of English Language

Please, check the suggestions on the title, abstract and Skinner's statements above.

Author Response

Dear Reviewer,

First of all, thank you for the suggestions. They were crucial to improving the essay's arguments, and I sought to incorporate all of them in the manuscript. All changes are highlighted in red in the reworked manuscript file and described below.

Reviewer 2

Reviewer: make explicit the notion of gender that underpins the discussion: the text clearly defines the behavioural framework used, and dialogues this framework with discussions from feminist fields. However, it does not state which notion of gender underlies its discussion (there are different conceptualizations of gender produced by different feminist strands). At different points in the text, gender seems to equate to "women", which is problematic. At some points, even, to "cisgender women". The concept and its discussion must be organized in a more explicit and referenced way, from the beginning of the text.

Reply: To avoid the problems indicated by the reviewer of reducing gender to women, I described and justified the focus at the end of the Introduction. In the Final Remarks, I indicated possibilities for future studies that would apply the theoretical proposal to contemplate other gender identities.

See (Introduction): “The focus is on the female gender and does not encompass other gender possibilities, such as male, non-binary, or fluid. Besides that, much of the following discussion about women's subjectivation is also within a binary, patriarchal, and cis-heteronormative gender system due to its predominance in Westernized society, as in the Brazilian one, and not because it is considered the only possible and desirable system (Laurenti 2023; Nicolodi and Hunziker, 2021; Pinheiro and Mizael 2019, 2023)”.

See (Final Remarks): “Future works could improve the reflections evoked by this study. For example, it is necessary to expand sources beyond the Brazilian behavior-analytic literature prioritized here. At first glance, the four theses on subjectivity could be used to understand other possibilities of gender and subjectivity not explored. However, the theoretical potential and limitations of this proposal still need to be assessed in a broader interpretative endeavor. For example, the assertions on subjectivity could be enriched with a discussion on intersex bodies, non-binary gender, and people with disabilities. Other axes of domination-exploitation, such as colonialism, and other social markers, such as age, religion, and class, must also be considered in further studies”.

Reviewer: Review the choice of the concept of patriarchy: the paper uses the concept of patriarchy to explain gender inequalities and sexism. However, this concept is not the most accurate to sustain the debate, since it conflicts with the theories and reflections proposed by black and intersectional feminisms. The idea of patriarchy assumes that gender oppression is the core of inequalities, while black feminists and intersectional feminists point out that sexism is one of the axes of oppression, and racism must be articulated with sexism, class prejudice, LGBTQIAphobia, and other oppressions. The use of these perspectives conflicts with the centrality of the concept of patriarchy in the paper, which must be reviewed.

Reply: I agree with the commentary about the centrality of patriarchy. To make the text more coherent, I used the expression "gender oppression" instead of "patriarchy" when appropriate. Besides that, I have sought to incorporate more of an intersectional lens in the manuscript.

Reviewer: The title of the article should be reformulated to address the rapprochement and contribution between the fields (radical behaviourism and feminist studies). The political meaning of the terms is discussed in the text, but does not seem to be the central issue of the work.

Reply: I agree with the assessment that the problem of political emptying of feminist terms was not the main point of the essay. Therefore, I completely changed the Title, Abstract, Introduction, and Final Remarks to accommodate the change in focus suggested by the reviewer.

Reviewer: The abstract should be revised. Up to line 12, it gives a generic justification of the article's problem, and the abstract ends up not including important definitions addressed in the work, such as the four theses developed on subjectivity.

Reply: The abstract has been completely remade and included the four theses on subjectivity

Reviewer: In the Introduction, on page 56, it is stated that "the objective is to explain the emancipatory potential of psychology from the perspective of Behavior Analysis.". However, this does not seem to be exactly the general objective of the paper. This passage should be revised;

Reply: The Introduction has been completely revised and the essay’s aim was also reformulated. These changes can be seen in the sixth paragraph of the Introduction (between lines 71 and 81).

Reviewer: On page 4, we read "Product of natural selection, such a body is a sensitive biochemical structure, being able to preserve or change its internal conditions depending on the exchange with the environment, maintaining organic balance (Skinner 1989).". It would be important to point out that there are biased approaches to behavioural psychology that use the evolutionary framework to explain and justify social practices, behaviours and human phenomena crossed by gender inequality. This sexist use of evolutionary theories is not addressed in the paper, and it would be important to point it out, especially as the author refers to natural selection in explaining what a body is for radical behaviourism.

Reply: I tried to rule out possible biological reductionist readings of the insertion of the bodily dimension in the understanding of subjectivity. Instead of dealing with the subject in a footnote, I addressed it in the second, third, and fourth paragraphs of item 2.2 Multidimensional Subjectivity.

Reviewer: Due to the lack of a definition of the concept of gender on which the article's discussion is based, at different points in the text it is not clear how the author understands the place of trans identities, especially trans women, in feminism. On page 8, in particular, the term identify (referring to trans people) is enclosed in quotation marks and the trans prefix is in italics. Why?

Reply: The proposal to contemplate other possibilities of gender and other subjectivities is exciting. But, exploring all these points would go beyond the scope of the text. As previously mentioned (Reply 1), I described and justified the focus on women’ subjectivation and female gender at the end of the Introduction. In the Final Remarks, I indicated possibilities for future studies that would apply the theoretical proposal to contemplate other gender identities.

Reviewer: Two passages, on page 7 (lines 315-317) and page 10 (lines 457-462), must be reviewed, in which it is not clear whether Skinner's statements refer to gender issues or individuals/concepts in general.

Reply: I added a paragraph that clarifies that I am doing the transposition of the discussion of subjectivity to gender issues. It was not covered by the behaviorist texts cited, including those by Skinner.

See (lines 104 to 107): Below, I present the theses, applying them to the constitution of women’s subjectivity – although the original behaviorist studies did not do so (including those of Skinner). Thus, I intend to outline an example of how this theory of subjectivity can approach feminist studies.

Reviewer: On page 10 there is a critique of depoliticizing approaches to the concept of "lugar de fala", which is only defined later in the text. It is necessary to define the concept here, to then criticize neoliberal interpretations of it.

Reply: As the essay’s focus is no longer on feminist terms, this concept was removed from the manuscript.

Reviewer: The terms "aversive techniques", "positive reinforcement" and “non-aversive control techniques” are behaviourist concepts that must be explained (maybe in a footnote) for non-specialist readers.

Reply: I reduced technical terms to avoid the need for explanatory notes for each of them, which would make the text too long. With this economy of technical terms, I intend to make the text more understandable for those who are unfamiliar with behaviorist expressions without having to rely heavily on explanatory notes.

Reviewer: Please, check the suggestions on the title, abstract and Skinner's statements above.

Reply: All these points were checked as previously described.

 I hope I have satisfactorily addressed the suggestions presented, and thank you once again for your valuable review.

Best regards.

Round 2

Reviewer 2 Report

Comments and Suggestions for Authors

The final version attended the suggestions and brings an important contribution for the field.